# Identifying and Overcoming Mechanisms of PARP Inhibitor Resistance in Homologous Recombination Repair-Deficient and Repair-Proficient High Grade Serous Ovarian Cancer Cells

**DOI:** 10.3390/cancers12061503

**Published:** 2020-06-09

**Authors:** Miriam K. Gomez, Giuditta Illuzzi, Carlota Colomer, Michael Churchman, Robert L. Hollis, Mark J. O’Connor, Charlie Gourley, Elisabetta Leo, David W. Melton

**Affiliations:** 1Nicola Murray Centre for Ovarian Cancer Research, Edinburgh Cancer Research UK Centre, MRC Institute of Genetics and Molecular Medicine, University of Edinburgh, Edinburgh EH4 2XU, UK; miriamkathleengmz1@gmail.com (M.K.G.); michael.churchman@igmm.ed.ac.uk (M.C.); Robb.Hollis@ed.ac.uk (R.L.H.); Charlie.Gourley@ed.ac.uk (C.G.); 2Early Oncology R&D, AstraZeneca, Cambridge CB4 0WG, UK; giuditta.illuzzi@astrazeneca.com (G.I.); ccolomer@neuraxpharm.com (C.C.); Mark.J.OConnor@astrazeneca.com (M.J.O.); elisabetta.leo@astrazeneca.com (E.L.)

**Keywords:** PARP inhibitor, olaparib, resistance mechanism, DNA repair, WEE1 kinase, ovarian cancer

## Abstract

High grade serous ovarian cancer (HGSOC) is a major cause of female cancer mortality. The approval of poly (ADP-ribose) polymerase (PARP) inhibitors for clinical use has greatly improved treatment options for patients with homologous recombination repair (HRR)-deficient HGSOC, although the development of PARP inhibitor resistance in some patients is revealing limitations to outcome. A proportion of patients with HRR-proficient cancers also benefit from PARP inhibitor therapy. Our aim is to compare mechanisms of resistance to the PARP inhibitor olaparib in these two main molecular categories of HGSOC and investigate a way to overcome resistance that we considered particularly suited to a cancer like HGSOC, where there is a very high incidence of *TP53* gene mutation, making HGSOC cells heavily reliant on the G2 checkpoint for repair of DNA damage and survival. We identified alterations in multiple factors involved in resistance to PARP inhibition in both HRR-proficient and -deficient cancers. The most frequent change was a major reduction in levels of poly (ADP-ribose) glycohydrolase (PARG), which would be expected to preserve a residual PARP1-initiated DNA damage response to DNA single-strand breaks. Other changes seen would be expected to boost levels of HRR of DNA double-strand breaks. Growth of all olaparib-resistant clones isolated could be controlled by WEE1 kinase inhibitor AZD1775, which inactivates the G2 checkpoint. Our work suggests that use of the WEE1 kinase inhibitor could be a realistic therapeutic option for patients that develop resistance to olaparib.

## 1. Introduction

Ovarian cancer is the eighth most common female cancer and the eighth most common cause of female cancer death worldwide (GLOBOCAN 2018 estimates, [1]). High grade serous ovarian cancer (HGSOC) comprises 75–80% of ovarian cancers and is characterized by *TP53* mutation and genetic heterogeneity [2]. The two major molecular categories are homologous recombination repair (HRR)-proficient and HRR-deficient, which is often due to the loss of function of the *BRCA1* or *BRCA2* genes [3]. Most HGSOC cases are treated with combined surgery and chemotherapy based on carboplatin and a taxane [4]. Although patients with HRR-deficient cancers initially respond well, the majority relapse after first line chemotherapy.

Poly (ADP-ribose) polymerase 1 (PARP1) is essential for the repair of DNA single-strand breaks (SSBs) [5]. PARP inhibitors (PARPi), such as olaparib, block the catalytic site on PARP1 and trap the enzyme on the DNA while preventing catalytic activity [6]. The structural basis for the ability of PARPi to trap PARP1 on the DNA has recently been determined [7]. Unrepaired SSBs with trapped PARP1 that persist into replication are believed to lead to the collapse of replication forks [8]. The resulting DNA double-strand breaks (DSBs) are preferentially and accurately repaired by HRR. In *BRCA*-deficient tumors, the HRR pathway is not functional and repair is directed instead toward the error-prone non-homologous end joining (NHEJ) pathway, resulting in large-scale genome instability and cell death. Thus, PARP inhibition is synthetically lethal in HRR-deficient HGSOC, with *BRCA*-deficient cells displaying ~1000-times greater sensitivity to PARP inhibitors than wild-type cells [9,10]. Clinical response to PARP inhibitor olaparib was first demonstrated in germline *BRCA* mutation carriers in phase 1 trials [11,12]. A phase 2 study reported that PARP inhibitors might be an alternative to conventional chemotherapy, with a more favorable toxicity profile, in a patient population with relapsed disease and *BRCA* mutation [13]. Another phase 2 study, in the relapsed HGSOC maintenance setting, showed prolonged progression-free survival (PFS) of 8.4 months in the olaparib arm, vs. 4.8 months in the placebo arm [14]. Recently, a landmark phase 3 trial (SOLO1) found that use of olaparib as first-line maintenance therapy provided a significant PFS benefit (risk of disease progression and death was 70% lower with olaparib than with placebo) among women with newly diagnosed advanced ovarian cancer and a *BRCA* mutation [15].

There is growing evidence that olaparib [16] and other PARPi may also be effective in the treatment of HGSOCs without clearly identifiable defects in the HRR pathway. A phase 3 trial of niraparib maintenance therapy in platinum-sensitive, recurrent HGSOC reported a > 3-fold increase in PFS over placebo for patients who had either germline *BRCA* mutations, or another cause of HRR deficiency. Niraparib also improved PFS for patients in the HRR-proficient subgroup [17]. A similar result was obtained in a phase 3 trial for another PARPi, rucaparib, being used in maintenance treatment for recurrent ovarian carcinoma. Rucaparib also improved PFS for patients in the HRR-proficient subgroup [18]. Recent analysis of tumour samples from an olaparib maintenance monotherapy phase 2 trial for platinum-sensitive, recurrent HGSOC (Study 19) has also reported an increase in PFS for the HRR-proficient subgroup [19]. In each of these three studies HRR-proficiency was defined as no *BRCA* mutations and no evidence of an HRR deficit in a next generation sequencing-based assay. The molecular basis for this unanticipated benefit in HRR-proficient HGSOC could be the ability of PARPi to trap PARP1 on replicating DNA, but PARP1 is also reported to be able to protect HRR from interference by proteins involved in NHEJ [20].

While PARP inhibitors have greatly improved treatment options for HRR-deficient HGSOC, development of PARP inhibitor resistance in some patients limits the clinical outcome [21]. In vitro and in vivo models suggest that PARPi resistance can occur by a variety of independent mechanisms [22,23,24,25,26,27,28,29,30,31,32,33,34,35,36]. The aim of this study is to compare mechanisms of resistance to PARP inhibitor olaparib in HRR-proficient and -deficient HGSOC cells and then use the resistant cells to test a method to overcome olaparib resistance considered particularly appropriate for HGSOC. Near universal *TP53* mutation in HGSOCs [3] inactivates the G1 checkpoint making them heavily reliant on the G2 checkpoint for DNA damage repair and survival. This checkpoint is controlled by the phosphorylation status of CDK1. WEE1 kinase responds to DNA damage with inhibitory phosphorylation of CDK1, causing G2 arrest and so giving time for DNA repair before entering into mitosis [37]. For this reason, we sought to test whether HGSOC cells with acquired resistance to olaparib could effectively respond to treatments with the WEE1 inhibitor AZD1775 [38].

## 2. Results

### 2.1. Generation of Olaparib-Resistant HGSOC Clones

From our panel of validated HGSOC cell lines [39,40], ES-2 (HRR-proficient) and OVCAR8 (HRR-deficient due to hypermethylation of the *BRCA1* gene promoter [41]) were chosen for the isolation of olaparib-resistant clones because of their good colony forming ability when seeded thinly. Independent olaparib-resistant ES-2 and OVCAR8 clones were isolated by thinly plating cells (1000 and 1500 cells per well) in a 96-well plate and treating with 25 or 50 µM olaparib for ES-2 and 12 µM olaparib for OVCAR8. These selective concentrations of olaparib were chosen based on the results of growth assays described in the next paragraph. Colonies became visible after four weeks and arose at a frequency of 8 × 10^−5^ for ES-2 and 5 × 10^−5^ for OVCAR8. Only wells containing single colonies were used to isolate resistant clones, which were expanded and maintained under continuous selection. Six clones resistant to 25 µM and two clones resistant to 50 µM olaparib were isolated from ES-2, while nine clones resistant to 12 µM olaparib were isolated from OVCAR8. Resistant clones used for further analysis are shown in Figure 1. The nomenclature system used for resistant clones was chosen to indicate the cell line used, selective agent, its concentration, and an identifying number. Thus, ES-2 OLA_50:1 is ES-2 clone number 1, selected for resistance to 50 µM olaparib.

The level of olaparib resistance in individual ES-2 and OVCAR8 clones was first determined using a standard five-day Sulphorhodamine B (SRB) growth assay [42]. The HRR-proficient ES-2 cell line had an IC50 for olaparib of 25 µM in the five-day SRB growth assay, while the HRR-deficient and so more sensitive OVCAR8 line had a 10-fold lower olaparib IC50 of 2 µM in the same assay (Figure 2A,B). All the four OVCAR8 clones assayed (IC50s ranging from 50–65 µM) showed > 25-fold increases in IC50 values compared to the OVCAR8 parent (IC50 2 µM; Figure 2B). Apparent levels of resistance in this assay were much less for the four ES-2 clones (IC50s ranging from 30–65 µM) compared to the ES-2 parent (IC50 25 µM): 2–3-fold higher for the 50 µM clones, and taking the 95% confidence intervals into account, at best only marginal increases for 25 µM clones (Figure 2A). Reasoning that the standard five-day growth assay was too short to reveal the true difference in olaparib resistance between our ES-2 clones and the ES-2 parent, because the phenotypic effects of PARPi require > 5 population doubling times for accurate measurement [43,44], we instead used an eleven-day survival assay (Figure 2C). In this survival assay the IC50 for the ES-2 parent (4 µM) was much lower than in the shorter growth assay (25 µM), while the IC50s for the resistant clones were essentially unaffected. The two clones resistant to 25 µM olaparib now showed ~10-fold increased IC50 values compared to the ES-2 parent, while the IC50 value for ES-2 OLA_50:1 was 20-fold higher, confirming the improved accuracy of a long term proliferation assay for the evaluation of PARPi.

### 2.2. Changes Associated with Olaparib Resistance in HGSOC Clones

To investigate the mechanism of olaparib resistance in our HRR-proficient and -deficient HGSOC clones, protein lysates were Western blotted for proteins previously found to be involved in PARPi resistance mechanisms in HRR-deficient cells [22,23,24,25,26,27,28,29,30,31,32,33,34,35,36]. Lysates were made from resistant clones grown under olaparib selection, while parental cell lysates were made from cells in ordinary medium. Blots for ES-2 and three resistant ES-2-derived clones are shown in Figure 3A and histograms showing the relative expression of the proteins in resistant clones, compared to the ES-2 parent, are shown in Figure 3B. PARP1 in the ES-2 parent showed a low level of self poly (ADP-ribosylation) [auto-PARylation]. Although PARP1 levels were unaltered in resistant clones, the ability of olaparib to block PARP1 auto-PARylation in the resistant clones, which were maintained under selection, was clearly shown. Some increment in levels of γH2AX, the chromatin marker for DSBs [45], was seen in all olaparib-resistant clones, most likely as a consequence of the olaparib treatment. There was no evidence for upregulation of the MDR1 membrane glycoprotein efflux pump being involved in resistance. The most noticeable change among proteins analyzed here, that are known to be involved in olaparib resistance, was the major reduction (> 5-fold) in levels of PARG (PAR glycohydrolase) in all the resistant clones. While REV7 levels were unaffected in resistant clones, all clones analyzed showed reduced levels of RIF1 (with a major reduction in clone ES-2 OLA_50:1) and 53BP1 (with a major reduction in clone ES-2 OLA_25:6). Although one of the two 25 µM clones (ES-2 OLA_25:5) showed a decreased level of EZH2 and H3K27Me3, levels of MUS81 endonuclease were largely unaffected in all clones.

Four OVCAR8-derived clones were also tested for changes in protein expression relevant to olaparib resistance (Figure 4). As with ES-2 clones, PARP1 levels were unaltered in OVCAR8-resistant clones and the ability of olaparib to block auto-PARylation of PARP1 was again evident in clones maintained in olaparib. Compared to ES-2 clones, there was consistently greater elevation of the γH2AX DNA damage marker. Again, as in ES-2, there was no evidence that increased levels of MDR1 were involved in resistance. The low level of BRCA1 protein observed in the OVCAR8 parent was expected, given that methylation of the *BRCA1* gene promoter has been reported in this cell line [41]. All resistant OVCAR8 clones showed upregulation of BRCA1 protein levels, with the largest increase in OVCAR8 OLA_12:3. Two of the clones with the smallest increase in BRCA1 protein (OVCAR8 OLA_12:4 and OVCAR8 OLA_12:5) also showed major reductions (> 10-fold) in PARG levels. The same two clones also showed equivalent large reductions in levels of RIF1 and modest reduction in 53BP1, while REV7 levels were unaffected. Clone OVCAR8 OLA_12:1 also showed a modest reduction in levels of RIF1. No changes previously associated with olaparib resistance were seen in the EZH2/H3K27Me3/MUS81 replication fork stabilization pathway. The elevated level of H3K27Me3 in clones OVCAR8 OLA_12:1 and OVCAR8 OLA_12:3 is the opposite of the change in this histone mark that has been associated with olaparib resistance and its significance is unclear.

Thus, different protein changes previously reported to cause PARPi resistance were seen in ES-2 and OVCAR8 clones, with examples in both cell types of multiple changes in the same clone.

### 2.3. Olaparib-Resistant HGSOC Clones Remain Sensitive to WEE1 Inhibition

The very high frequency of *TP53* mutation in HGSOC makes this cancer heavily dependent on the G2 checkpoint, controlled by WEE1 kinase, for cell cycle arrest and DNA damage repair. We next investigated whether the growth of our olaparib-resistant ES-2 and OVCAR8 clones could be controlled by WEE1 kinase inhibitor AZD1775 in our five-day SRB growth assay (Figure 5). Of the three ES-2 clones tested, only ES-2 OLA_50:1 showed a small increase (< 2-fold) in AZD1775 IC50 value compared to ES-2 (Figure 5A). While all four OVCAR8 clones had the same sensitivity to AZD1775 as the parent OVCAR8 cell line (Figure 5B). We conclude that clones from these two HGSOC cell lines that display resistance to olaparib remain as sensitive to AZD1775 as their parental cell lines.

## 3. Discussion

The detection of SSBs by PARP1 triggers a DNA damage response (DDR) leading to their repair [46]. PARP inhibition is particularly effective in one of the two major molecular categories of HGSOC, where the HRR pathway for DSBs is defective [11,12,13,14,15]. As a consequence of PARP inhibition and PARP1 trapping on the DNA, SSBs persist into replication resulting in their conversion to DSBs. The ability of PARPi to trap PARP1 on replicating DNA [6] and a report that PARP1 can protect HRR from interference by proteins involved in NHEJ [20], raise the possibility that it could also be effective in some cases of HRR-proficient HGSOC. This suggestion has been supported by analysis of three recent trials of maintenance therapy with PARP inhibitors olaparib, niraparib, and rucaparib in platinum-sensitive, recurrent HGSOC. In addition to major survival benefits in HRR-deficient patients, there were also increases in PFS for patients in the HRR-proficient subgroups [17,18,19].

To further explore this preclinically, we chose to isolate olaparib-resistant clones from both HRR-deficient (OVCAR8) and HRR-proficient (ES-2) HGSOC cell lines, so that we could investigate resistance mechanisms and test ways to overcome olaparib resistance, particularly by exploiting the *TP53* gene deficiency found in HGSOC.

A previous study of patient-derived HGSOC xenografts found that methylation of all *BRCA1* gene copies, leading to silencing of the gene, predicted sensitivity to PARPi, while heterozygous methylation was associated with resistance [47]. In the HRR-proficient ES-2 cell line, the IC50 for olaparib in a 5-day growth assay was > 10-fold higher than in OVCAR8, where methylation of the *BRCA1* gene promoter has been reported [41] and is the likely cause of the HRR-deficiency [48] and resulting sensitivity to olaparib and other PARPi. Olaparib-resistant OVCAR8 clones had > 25-fold higher IC50s than the OVCAR8 parent in the 5-day growth assay, whereas olaparib-resistant ES-2 clones showed a much smaller IC50 increase (< 3-fold) over the ES-2 parent (IC50 25 µM) in the same assay. When an 11-day survival assay was used instead, to give the longer time needed for PARP inhibitor-treated cells with an intact HRR pathway to go through enough cell doublings to accumulate sufficient DNA damage to result in cell death [43,44], the difference between the ES-2 parent (IC50 now reduced to 4 µM) and resistant clones became much more apparent, with IC50s up to 20-fold higher than the parent.

Multiple mechanisms of PARPi resistance have been described, mostly in HRR-deficient breast cancer or HGSOC cells with *BRCA* mutations. Changes that preserve the DNA damage response (DDR) triggered by the action of PARP1 at SSBs, such as altered expression or mutation of *PARP1* [22], or downregulation of the enzyme PARG (poly [ADP-ribose] glycohydrolase), that removes PARP-catalyzed poly (ADP-ribosylation) [PARylation], can both result in resistance [23]. Restoration of HRR function by secondary *BRCA* mutation [24,25] is the only clinically confirmed resistance mechanism reported so far. Translation initiation downstream of a frameshift mutation [26], or stabilization of mutant BRCA proteins [27], can also lead to resistance. Enhanced drug efflux, resulting from increased expression of the MDR1/PgP (multidrug resistance protein 1/P-glycoprotein) membrane transporter protein, was also linked to acquired resistance in preclinical models [28,29]. Reduced expression of 53BP1 (p53 binding protein 1) in *BRCA1*-deficient cells, or of members of its effector complex, such as RIF1 and REV7, which normally act to prevent DSBs from undergoing BRCA1-mediated end processing and instead direct repair down the error-prone NHEJ pathway, can boost HRR activity resulting in PARPi resistance [30,31,32,33]. Stabilization of stalled DNA replication forks by inhibition of MRE11 nuclease in *BRCA*-deficient cells can also lead to PARPi resistance [34]. Localization of the EZH2 protein at stalled replication forks trimethylates Lys27 on Histone H3 and so recruits MUS81 nuclease to degrade the fork. Low EZH2 levels prevent MUS81 recruitment and so stabilize the fork, resulting in resistance to PARPi [35]. Activation of epithelial-mesenchymal transition in a mouse model of *BRCA2*-deficient breast cancer has also been found to trigger olaparib resistance [36].

Here we have investigated mechanisms of olaparib resistance in HRR-deficient and -proficient HGSOC cells. We chose to use OVCAR8, with methylation of the *BRCA1* gene promoter as our HRR-deficient and platinum-sensitive cell line, rather than a *BRCA* mutant line from our HGSOC cell line panel, because of its superior ability to plate at the low cell densities required to isolate individual resistant clones. A major reduction in levels of PARG protein was found in all olaparib-resistant clones isolated from HRR-proficient ES-2 cells. The reduction was greatest in ES-2 OLA_50:1, the clone with the highest IC50 for olaparib in our survival assay. PARG degrades the PAR chains that are added by PARP1 at sites of SSBs and which are reported to be essential for DNA repair protein recruitment and processing of DNA damage. This reduction in PARG would be expected to preserve any residual PARylation carried out by PARP1 in the presence of olaparib and so rescue some downstream PARP signaling and repair of SSBs [23]. Reductions in the level of 53BP1 (especially in ES-2 OLA_25:6) and of RIF1 (especially in ES-2 OLA_50:1) in resistant ES-2 clones would direct repair of DSBs away from the error-prone NHEJ pathway and instead down the HRR pathway to lead to cell survival [30,31,32,33]. Levels of MUS81 were also lowest in ES-2 OLA_50:1, which could also aid olaparib survival by preserving the stability of stalled replication forks [35]. Unsurprisingly, more changes appeared to be necessary to achieve olaparib resistance in individual clones in the HRR-deficient OVCAR8 cells. Increased levels of BRCA1 were seen in all resistant clones which would be expected to boost HRR, with highest levels in OVCAR8 OLA_12:3. Major reductions in PARG and RIF1 and reduced levels of 53BP1 were also seen in OVCAR8 OLA_12:4 and OVCAR8 OLA_12:5, while the clones with the highest levels of BRCA1, OVCAR8 OLA_12:1, and OVCAR8 OLA_12:3, had smaller reductions in PARG, no reduction of 53BP1 and only OVCAR8 OLA_12:1 had reduced RIF1. There was no evidence in either cell line for a role in resistance of increased expression of the MDR1/PgP membrane transporter protein, which has been reported as a frequent cause of resistance to PARP inhibition in other studies in breast and ovarian cancer cells [28,29].

In these two different HGSOC cell models, we have observed several previously described olaparib resistance mechanisms. Independent of HRR status, PARG downregulation to preserve the DDR to SSBs was observed in most clones. A variety of different ways to boost HRR of DSBs were also observed, notably increased levels of BRCA1 protein and downregulation of 53BP1 and RIF1. No information was obtained during the study to explain the basis for these altered protein levels. Several of these resistance mechanisms co-existed in the same resistant cell clone, suggesting that in some cases a single mechanism may be insufficient for cells to overcome the actions of olaparib. As far as we are aware this is the first time that these mechanisms have been reported to occur together in the same resistant clone. Although functional studies would be needed to investigate this further, this result could be very relevant for olaparib resistance in the clinic, where it would suggest that more than one resistance mechanism may need to be addressed in the same tumour.

The ES-2 cell line is homologous recombination repair (HRR) proficient and so is naturally less sensitive to olaparib than the other ovarian cancer cell line used, OVCAR8, which is HRR-deficient (IC50 in our growth assay: ES-2, 25 µM; OVCAR8, 2 µM). We chose to isolate and analyze olaparib-resistant clones from both HRR-deficient and -proficient cell lines because of recent studies showing that patients with each molecular subtype of ovarian cancer can benefit from PARP inhibitor therapy [17,18,19]. The high concentrations of olaparib needed to isolate resistant clones from ES-2 cells could potentially impact other cell pathways. However, our observation that similar changes associated with olaparib resistance were seen in resistant clones from both cell types suggests that this is not the case.

With the increasing use of PARPi maintenance therapy for HGSOC patients with HRR-deficiency and now also for HRR-proficient HGSOC as well, the inevitable development of resistance will pose a major challenge, necessitating the use of an alternative therapeutic strategy to control tumour growth. AZD1775 (formerly known as MK-1775) was the first reported WEE1 kinase inhibitor [49]. It is a highly selective, potent, ATP-competitive, small molecule inhibitor of the kinase domain with an enzyme IC50 of 5.2 nM in a cell-free assay. Small cell lung cancer patient-derived circulating tumour cell explant models with HRR-deficiency were found to respond well to olaparib in combination with WEE1 inhibitor AZD1775 [50]. The very high frequency of *TP53* mutation in HGSOC [2] makes this cancer heavily dependent on the G2 checkpoint that is controlled by WEE1 kinase for cell cycle arrest and DNA damage repair. Simultaneous treatment of ovarian cancer xenograft models with PARP and WEE1 inhibition was effective but poorly tolerated, while sequential administration retained efficacy with reduced toxicity [51]. In this study we found that olaparib-resistant ES-2 and OVCAR8 clones remained sensitive to AZD1775, suggesting that use of WEE1 kinase inhibition for patients developing resistance to olaparib could also be a realistic therapeutic option.

## 4. Materials and Methods

### 4.1. Mammalian Cell Culture Assays

Human HGSOC cell lines ES-2 and OVCAR8 were obtained from the American Type Culture Collection, where they were authenticated by short tandem repeat profiling. Both cell lines are *TP53* deficient: the HRR-proficient ES-2 cell line has a *TP53* missense mutation (S241F, rs28934573), while the HRR-deficient and platinum-sensitive OVCAR8 cell line has a pathogenic *TP53* splice site SNP and also methylation of the *BRCA1* gene promoter [52]. All experiments with these cells were carried out within 10 passages of supply. Cells were cultured in DMEM (ES-2; #41965, Thermo Fisher Scientific, Paisley, UK), or RPMI-1640 medium (OVCAR8; #21875, Thermo Fisher Scientific), supplemented with 10% FCS, non-essential amino acids (#11140-035, Thermo Fisher Scientific), 1 mM sodium pyruvate, 2 mM L-glutamine, 0.01 mg/mL insulin and penicillin (100 U/mL)—streptomycin (100 mg/mL) at 37 °C, 5% CO_2_.

To isolate clones resistant to olaparib (S1060, Selleckchem, Houston, TX, USA), cells were plated at 1000 and 1500 cells/well in 96-well plates in medium containing the selective olaparib concentration (25 or 50 µM for ES-2 and 12 µM for OVCAR8, determined from the results of 5-day growth assays). Selection for olaparib resistance was carried out in 96-well plates. To ensure that our analysis was carried out on single olaparib-resistant clones, rather than clonal mixtures, the plating density was adjusted so that the average number of resistant colonies arising/well was < 1. Wells were examined microscopically and colonies were only trypsinized from wells that clearly contained just a single colony. Colonies typically went through just four passages under continuous olaparib selection (24-well plate, 6-well plate, 25 cm^2^ flask, 25 cm^2^ flasks) before sufficient cells were available to make protein lysates for analysis.

Sensitivity of cell lines to olaparib, or WEE1 inhibitor AZD1775 (S1525, Selleckchem), was determined by a five-day Sulphorhodamine B (SRB) growth assay [41]. Cells were plated at 2000 cells per well into 96-well plates containing an olaparib or AZD1775 dilution series (8 wells for each drug concentration). Dose response curves and IC50 values, with 95% confidence intervals, were obtained with GraphPad Prism (San Diego, CA, USA) using a non-linear regression curve fit model.

A clonogenic survival assay was also used on olaparib-resistant ES-2 clones. Cells were plated at 100,000 cells per well in 6-well plates in medium containing an olaparib dilution series and left for 11 days to allow surviving cells to form colonies. Plates were then fixed, stained, and processed in the same way as for the SRB growth assay.

### 4.2. Western Blotting

Protein extraction was carried out on ice using RIPA buffer (25 mM Tris-HCl pH 7.2, 150 mM NaCl, 1% Triton X-100, 1% deoxycholate, 1 mM EDTA, 20 mM NaF, 100 µM orthovanadate), with Halt protease and phosphatase inhibitor cocktail (Thermo Fisher Scientific, #78430). Proteins were quantified with BCA protein assay kit (Thermo Fisher Scientific, #23225). Total of 50 µg of protein lysates were loaded in each lane of NuPAGE Novex 4–12% bis-tris midi protein gels (Thermo Fisher Scientific, #WG1402A) and proteins were transferred onto nitrocellulose membrane using the iBlot^®^ 2 Dry Blotting System (Thermo Fisher Scientific). Membranes were cut, blocked in TBS-Tween + 5% milk, and then incubated overnight at 4 °C with the primary antibodies as listed in Appendix A. After washes in TBS-Tween, membranes were incubated with the respective secondary antibodies: HRP-conjugated, anti-mouse (#7076S, Cell Signaling Technology, Danvers, MA, USA), or anti-rabbit (#7074S, Cell Signaling Technology). Membrane chemiluminescence was developed with Pierce West Dura substrate (Thermo Fisher Scientific, #34080) and acquired using GBOX (Syngene, Cambridge, UK). Protein band quantification was performed with ImageJ and normalized on the GAPDH signal used as loading control.

## 5. Conclusions

We have identified multiple changes involved in resistance to PARP inhibition in homologous recombination repair (HRR)-deficient and also in HRR-proficient high-grade serous ovarian cancer (HGSOC) cells. The most frequent change, involving major reduction in levels of poly (ADP-ribose) glycohydrolase, would be expected to preserve a residual DNA damage response initiated by PARP1 in response to DNA single-strand breaks. Other changes seen would be expected to boost levels of HRR of DNA double-strand breaks. Different resistance mechanisms co-existed in the same resistant cell clone, suggesting that, in some circumstances, a single mechanism may be insufficient for cells to overcome the actions of olaparib. As far as we are aware, this is the first time that these mechanisms have been reported to occur together in the same resistant clone. In a strategy that exploits the *TP53* deficiency found in HGSOC, growth of all olaparib-resistant clones could be controlled by WEE1 kinase inhibitor AZD1775.

## Figures and Tables

**Figure 1 cancers-12-01503-f001:**
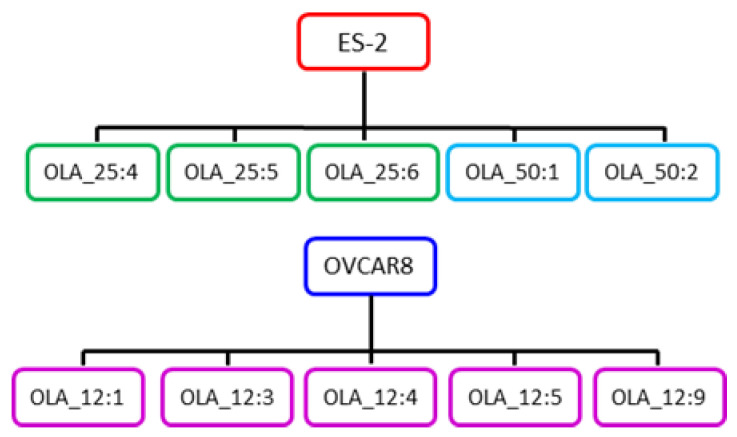
Olaparib-resistant high grade serous ovarian cancer (HGSOC) clones. Independent clones resistant to olaparib were first isolated from the homologous recombination repair (HRR)-proficient ES-2 cell line. Clones shown in green and blue boxes were isolated after parental cells were challenged with 25 and 50 µM olaparib, respectively. Olaparib-resistant clones were also isolated from the HRR-deficient OVCAR8 cell line. Clones shown in pink boxes were isolated after parental cells were treated with 12 µM olaparib. All the resistant clones were isolated in a single selection step from their ES-2, or OVCAR8 parent. Only clones used for subsequent analysis are shown here.

**Figure 2 cancers-12-01503-f002:**
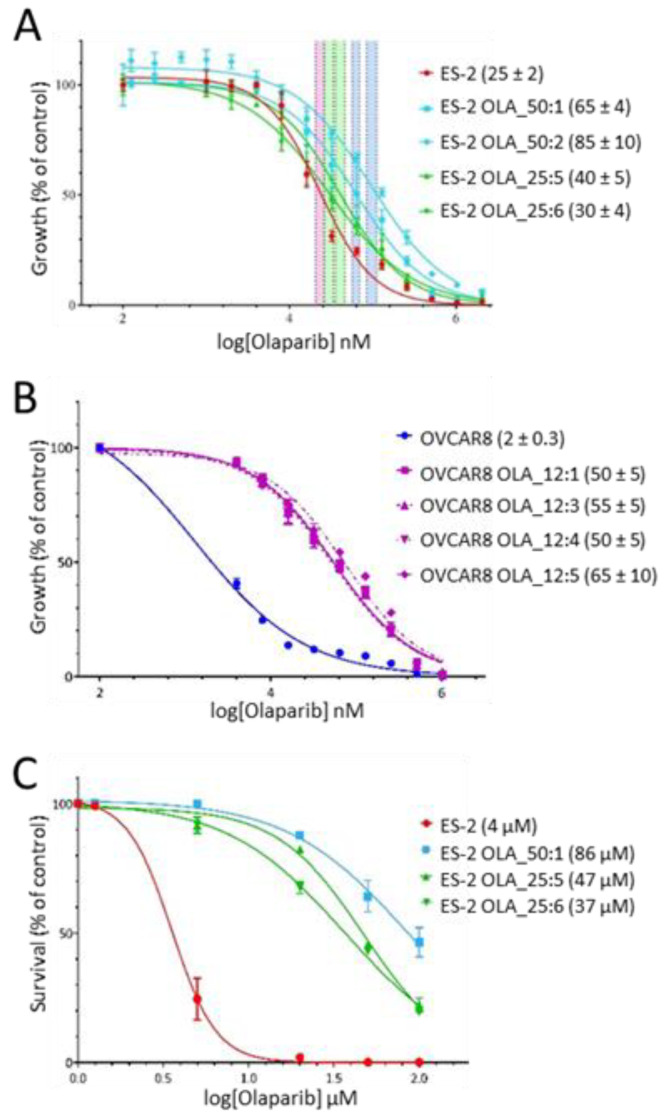
Decreased sensitivity of olaparib-resistant clones to olaparib. (**A**) Five-day growth assay for four ES-2 clones resistant to 25 or 50 µM olaparib. ES-2 parent is shown in red, 25 µM clones are in green, and 50 µM clones are in blue. IC50 values for each cell line (µM ± SEM) are shown in parentheses. 95% confidence intervals for IC50 values are shown as colored vertical bars. (**B**) Five-day growth assay for four OVCAR8 clones resistant to 12 µM olaparib. OVCAR8 parent is shown in blue, resistant clones are in pink. IC50 values (µM ± SEM) are shown in parentheses. (**C**) Eleven-day survival assay for three of the ES-2 clones shown in the growth assay above. ES-2 parent in red, 25 µM clones in green, 50 µM clone in blue. IC50 values (µM) are shown in parentheses.

**Figure 3 cancers-12-01503-f003:**
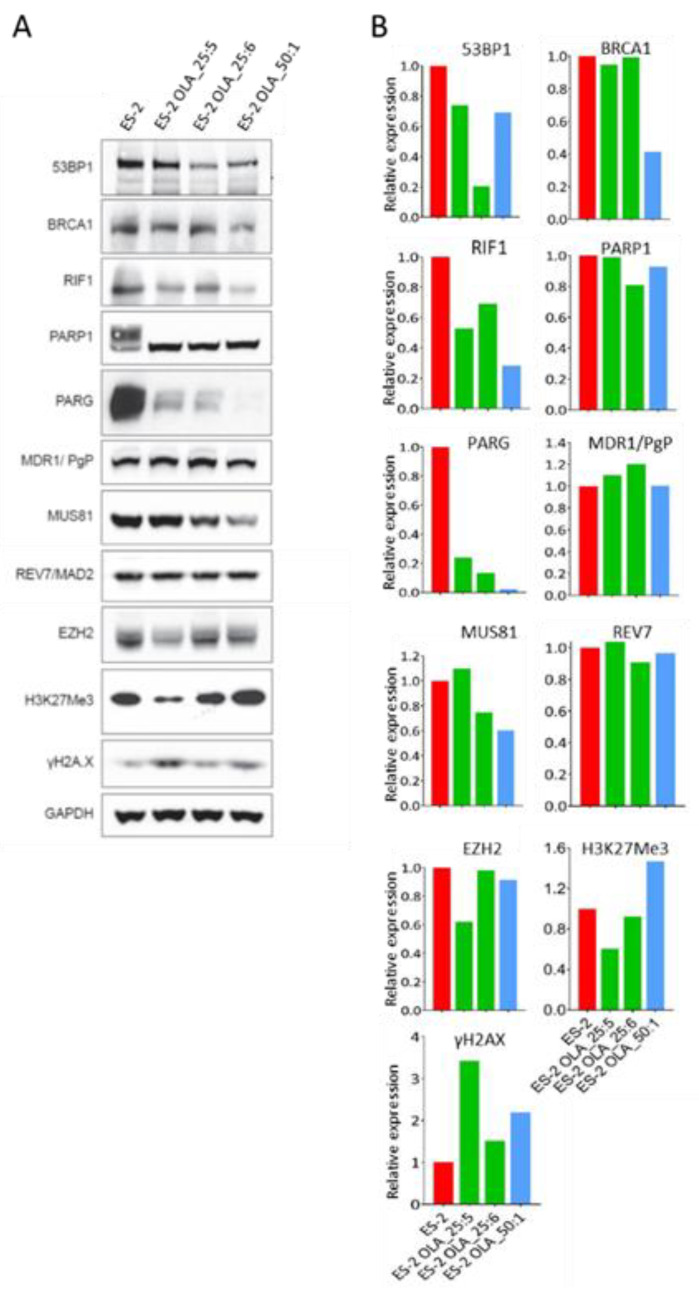
Changes in protein levels in ES-2 derived olaparib-resistant clones. Protein lysates were made from the ES-2 parent grown in normal medium and ES-2 clones maintained continuously under olaparib selection. (**A**) Western blots for 53BP1 (mol. wt. 214 kDa), BRCA1 (220 kDa), RIF1 (265 kDa), PARP1 (116 kDa), PARG (111 kDa), MDR1 (170 kDa), MUS81 (62 kDa), REV7 (24 kDa), EZH2 (85 kDa), H3K27Me3 (15 kDa), γH2AX (14 kDa) from ES-2 and three olaparib-resistant ES-2 derived clones, with GAPDH (37 kDa) as loading control. (**B**) Histograms showing levels of the different proteins in each clone, corrected for differences in loading control, and expressed relative to the level in the ES-2 parent. ES-2 parent shown in red. ES-2 OLA_25:5 and ES-2 OLA_25:6, resistant to 25 µM olaparib, are shown in green. ES-2 OLA_50:1, shown in blue, is resistant to 50 µM AZD1775. Note that the apparent reduction in levels of BRCA1 in clone ES-2 OLA_50:1 most likely reflects a problem with the transfer. Information supporting the identification of proteins in these western blots is shown in Appendix A.

**Figure 4 cancers-12-01503-f004:**
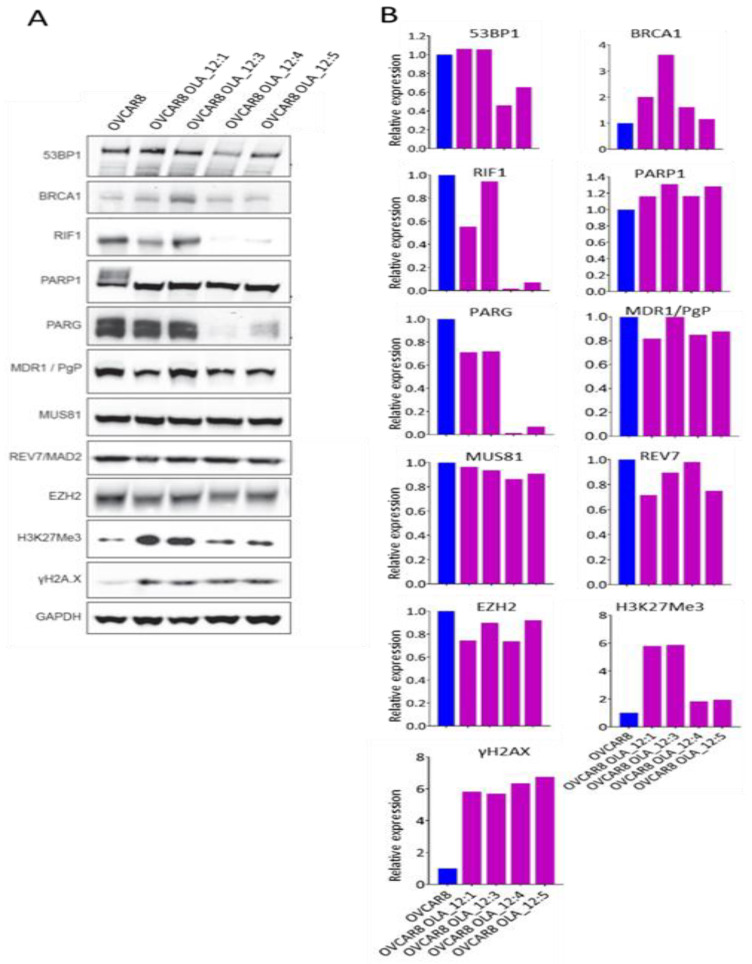
Changes in protein levels in OVCAR8-derived olaparib-resistant clones. Protein lysates were made from the OVCAR8 parent grown in normal medium and OVCAR8 clones maintained continuously under olaparib selection. (**A**) Western blots for 53BP1, BRCA1, RIF1, PARP1, PARG, MDR1, MUS81, REV7, EZH2, H3K27Me3, γH2AX from OVCAR8 and four olaparib-resistant OVCAR8 derived clones, with GAPDH as loading control. (**B**) Histograms showing levels of the different proteins in each clone, corrected for differences in loading control, and expressed relative to the level in the OVCAR8 parent. OVCAR8 parent shown in blue. OVCAR8 clones OVCAR8 OLA_12:1, OVCAR8 OLA_12:3, OVCAR8 OLA_12:4, and OVCAR8 OLA_12:5, resistant to 12.5 µM olaparib, are shown in dark pink. Information supporting the identification of proteins in these Western blots is shown in Appendix A.

**Figure 5 cancers-12-01503-f005:**
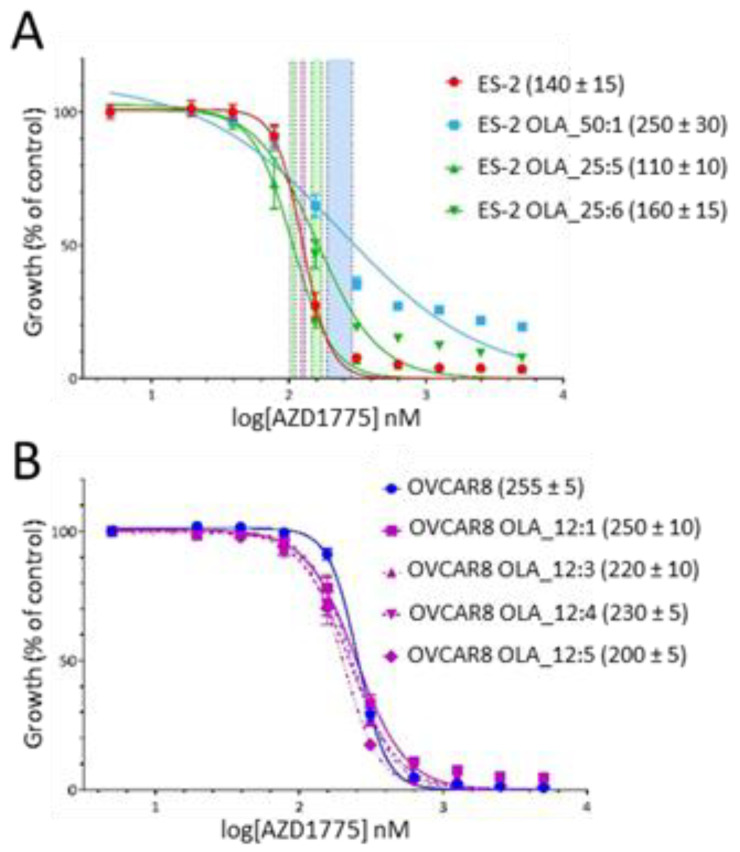
Sensitivity of olaparib-resistant HGSOC clones to WEE1 inhibition. (**A**) Olaparib resistant ES-2 clones remain largely sensitive to AZD1775. AZD1775 growth curves for three independent ES-2 clones resistant to 25 µM (green) or 50 µM olaparib (blue) are shown. ES-2 parent curve is in red. AZD1775 IC50 values for each curve (in nM ± SEM) are shown in brackets. 95% confidence intervals for IC50 values are shown as colored vertical bars. (**B**) Olaparib resistant OVCAR8 clones remain sensitive to AZD1775. AZD1775 growth curves for four OVCAR8 clones resistant to 12 µM olaparib (pink) are shown. OVCAR8 parent curve is in blue. AZD1775 IC50 values for each curve (in nM ± SEM) are shown in brackets. All olaparib-resistant clones show the same sensitivity to AZD1775 as the OVCAR8 parent.

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
