# Peer review of "Identifying and Overcoming Mechanisms of PARP Inhibitor Resistance in Homologous Recombination Repair-Deficient and Repair-Proficient High Grade Serous Ovarian Cancer Cells"

_cancers, 2020, doi:10.3390/cancers12061503_

Round 1
Reviewer 1 Report
This manuscript by Gomez et al., shows mechanisms of resistance to treatment with olaparib in two lines of high grade serous ovarian cancer (HGSOC), one deficient in HR and the other HR-proficient. Once the clones resistant to treatment with olaparib have been established, they demonstrate that these clones are sensitive to treatment with Wee1 inhibitors. The authors confirm the recent proposal (Fang Y et al., Cancer Cell. 2019) for the sequential use of PARPi and Wee1 inhibitors for the treatment of ovarian cancer.
1.- The authors conclude that different resistance mechanisms co-existed in the same resistant cell clone, suggesting that, in some circumstances, a single mechanism may be insufficient for cells to overcome the actions of Olaparib. How can authors be sure that they have not a mixture of cells rather than multiple mutations within the same cell? How many passes have the cells received during expansion until they are tested?
2.- Pag 2, line 45: Please, specify which PARP family member is required for the repair of ssDNA breaks.
3.- When the authors referred to PARP enzyme along the manuscript, in fact to which PARP family member they are referring? Please, specify.
4.- The use of a dose of 50 uM of olaparib for ES-2, seems to really indicate that this cell line is resistant to olaparib. These high concentrations of olaparib may impact other cell pathways?
5.- References are citated in the text as number but listed without indicating any number in the citation list.
Reviewer 2 Report
Identifying and overcoming mechanisms of PARP inhibitor resistance in homologous recombination repair-deficient and repair-proficient high grade serous ovarian cancer cells
The authors demonstrated that the decrease of PARG
The information about PARP inhibitor in the introduction would be less informative. The content in the introduction could be shortened. Instead, upfront recent studies regarding to iPARP resistance would be more informative to the readers.
Instead, introduction of Wee1 in cell cycle and inhibitor of this molecule would be informative.
It is unclear why figure 1 and figure 2 were displayed separately. These figures should be combined in one figure with separate panels.
Figure quality is very poor so that legend in the figures are not readable.
Basically, content in figure 3 and 4 is identical. Figures in figure 3 and 4 needs to be combined.
Presentation of altered protein levels shown in figure 3 and 4 is less meaningful. It is not clear why these proteins were selected.
No rationale was provided why Wee1 kinase inhibitor could sensitizes these cells.
Sensitivity of Wee1 kinase inhibitor to olaparib resistant cancer cells has been well-established. It is not clear what would be novel point of this study.
Results in the figures are very preliminary and lack clear statistical analysis. Not clear what would be novelty of this study.
No rationale was provided to use Wee1 kinase inhibitor based on data provided.
No scientific insight for the altered protein level was provided.
Overall, this study is very preliminary and descriptive without any novelty. They just provided the altered protein levels in olaparib resistant cancer cell lines without any scientific significance and provided one very preliminary sensitivity data that are already reported elsewhere.
Reviewer 3 Report
This study on PARP inhibitor resistance mechanism is quite interesting. The cells lines used are somewhat confusing thus a flow chart of the various ovary cancer Cell lines developed should included in this manuscript The signal observed for PARG is very strong usually the signal is weak and should show 3 bands. Thus it may be of some advantage to use gradient gels which are more resolutive. Finally some mechanistic explanation for the wee 1 Inhbitors should be included.The seminal paper of John Pascal and Ben Black in SCIENCE explaining in mechanistic terms the trapping mechanism should be added in the references.Author Response
please see attachment

Round 2
Reviewer 2 Report
No additional data were provided to solidify their finding.
The authors showed the following.
- Establishment of two cancer cell lines showing olaparib resistance
- Immunoblotting of a few proteins that are already known to be altered.
- Sensitivity of Wee1 inhibitor that has been already reported.
I could not still see the novelty of the manuscript despite the comment from the authors.
The data presented in the manuscript are still very preliminary and needs to be further supported by other approaches.
Reviewer 3 Report
The paper is acceptable for publication.